# Dancing with Giants: A Unified Framework for Cooperation Networks, Speed of Internationalisation, and Performance

**Telma Mendes** [1,2,*], **Carina Silva** [1] **and Alexandra Braga** [1]

1. Department of Business Science, Centro de Inovação e Investigação em Ciências Empresariais e Sistemas de Informação (CIICESI), Escola Superior de Tecnologia e Gestão (ESTG), Polytechnic Institute of Porto, 4610-156 Felgueiras, Portugal

2. Department of Business Organisation and Marketing, University of Vigo, 36310 Vigo, Spain

* Correspondence: tilm@estg.ipp.pt

**Abstract:** This study aims to explore whether the speed of internationalisation—measured by the change in international scale and the change in international scope—can mediate the relationship between network clustering (cooperation networks) and clustered firms' performance. A quantitative methodology was used to accomplish this purpose, and the research model was tested using the Partial Least Squares Structural Equation Modelling (PLS-SEM). Based on a sample of 1491 Portuguese firms, this paper showed that network clustering directly and positively impacts clustered firms' performance. When considering the effect of the speed of internationalisation, the study revealed that network clustering also has an indirect, mediated impact on firms' performance, through the change in international scope (geographical diversification). Conversely, when accounting for the mediation of the change in international scale (degree of internationalisation), the results confirmed that this dimension of speed does not mediate the relationship between network clustering and firms' performance. This evidence, therefore, stresses the role of geographical diversification in shaping how well the clustered firms perform based on the networks established through industrial clusters.

**Keywords:** networks; clustered firms; industrial clusters; speed of internationalisation; performance

## 1. Introduction

In this era of globalisation characterised by rapid changes in the business environment, the speed of internationalisation has become an important issue in cross-broader development [1]. The increase in market integration has forced many firms to expand their businesses overseas to remain competitive. Thus, decisions about the speed of international expansion have become increasingly relevant, in terms of gaining and sustaining competitive advantage [1], and might influence resource allocation, performance and a firm's survival [2].

From a theoretical perspective, the term speed of internationalisation is essential for understanding the dynamics of foreign expansion and a firm's behaviour over time; however, there are limitations on how previous research has defined and measured speed [3]. According to Chetty, Johanson, and Martín Martín [3], the general conceptualization of speed implies a limited temporal view, since it only considers the time between the firm's inception and the first internationalisation, overlooking the subsequent period once the first international market is the achieved. Moreover, referring to speed only as time—the time that takes to internationalise—discards the central aspects of a firm's internationalisation, such as market knowledge and commitment [3].

The firms that develop their activities abroad operate in an unfamiliar environment. In this way, network relationships have been recognised as mechanisms with a strong influence on successful internationalisation processes [4,5]. In the case of industrial clusters, such networks provide trust between their members, allowing them to overcome several uncertainties and complex conflicts [6]. The physical closeness of clustered organizations

facilitates the establishment of various interactions and the behaviour of each entity is influenced by other agents in the cluster [7]. Several businesses within industrial clusters have been focused on network platforms rather than traditional face-to-face contacts to develop their activities [8].

The powerful instruments that explore the entire network of an industrial cluster remain relatively unexplored [9–12] and the research, to date, has failed to provide clear guidance on how specific network interactions influence foreign expansion [13,14]. In the literature, there is a general assumption that the establishment of network relationships has a positive effect on the speed of internationalisation [4,13,15–17]. Nonetheless, a recent research stream has claimed that such interactions do not influence speed [18–20].

Additionally, the firms that become international, at some point, must deal with higher competitive pressure. Then, managers should address the following question: how many resources should they allocate to explore international opportunities, to improve a firm's competitive advantage? The International Business (IB) theories characterise a firm's internationalisation as the ability to exploit competitive advantages and the desire to explore resources that strengthen organisational competitiveness and long-term performance [21].

Several scholars have suggested that a higher speed of internationalisation enhances a firm's performance [22–26]. However, fast-paced international expansion is not risk-free, and companies have no guarantee that this strategy will lead to better performance [1]. For these reasons, empirical research has reported mixed results about the relationship between the speed of internationalisation and performance, ranging from negative [27], positive [19,23,26], nonlinear [28–31], and even non-significant effects [2,32].

Thus, all these studies have improved our understanding of the relationship between cooperation networks, internationalisation, and performance. None of the studies have nevertheless fully documented the mechanisms by which clustered firms can use their networks to be internationally successful and, consequently, improve their performance. Several authors [9–14] have highlighted the lack of empirical studies assessing how variations in the speed of internationalisation affect not only their ability to benefit from the knowledge and learning provided by the clustered networks but also their performance, therefore calling for the development of theory in this field. As variations on the speed of internationalisation, as well as the level of organisational experience on foreign markets are likely to have a bearing on clustered firms' performance, several authors have called for the incorporation of this temporal dimension into performance models [1,19,23,26]. In addition, most of the research on cooperation networks and performance, particularly, measuring international scale [15,17,23,33,34] and international scope [34,35], is based on indicators reported to a certain fixed, static point of the time, accounting for the level of internationalisation in the firm, rather than its speed of internationalisation. It is the variability of these dimensions that reflect the speed of internationalisation expressed by the change in international scale and scope at two different moments, or the average change in both dimensions.

Here lies the problem statement of this paper, which beyond making explicit the differences between the level of internationalisation (time-invariant measure) and speed of internationalisation (time-variant measure), tries to explore in depth how the speed of internationalisation (measured by the change international scale and scope) can play a mediating role in the relationship between network clustering (cooperation networks) and clustered firms' performance. We particularly emphasise the role of the "speed of internationalisation" which is characterised by providing learning advantages, a high degree of specialised knowledge and the opportunity to occupy key positions in clustered networks both for creating value (due to their ability for knowledge exploitation) and for influencing market decisions (such as link science, institutions, industry, markets, and society).

The empirical analysis was carried out on a sample of 1491 Portuguese firms obtained from the Community Innovation Survey (The Community Innovation Survey (CIS) is the reference survey on innovation in enterprises. The European Union (EU) member

states first introduced the survey in 1992 and since then it has become the regular biennial data collection. At present, the survey is carried out in the EU, European Free Trade Association (EFTA), and the EU candidate countries. The legal framework for CIS since 2012 is the Commission Regulation No 995/2012 which establishes the quality conditions and identifies the obligatory cross-coverage of economic sectors, size class of enterprises, and innovation indicators. To comply with the Regulation requirements and to respond to the needs of several users, Eurostat together with the countries develops a standard questionnaire for each round—Harmonised Data Collection (HDC). Our study is drawn from the Portuguese version of the survey that contains data for the timeframe between 2012 and 2014. For more information on CIS, please refer to: https://ec.europa.eu/eurostat/web/microdata/community-innovation-survey (accessed on 25 July 2022)) [36], for the timeframe between 2012 and 2014. Portugal represents a particularly suitable setting for our study because it accounts for 19 industrial clusters geographically dispersed in the national territory [37], where most of its small and medium-sized enterprises (SMEs) display a significant share of exports [38].

In the remaining paper, we review the literature on the concepts under analysis, exploring the relationship between network clustering, speed of internationalisation, and performance. The following section describes the sample, data collection and the variables that have been used. Subsequently, we discuss the results processed by the Partial Least Squares Structural Equation Modelling (PLS-SEM). Finally, we introduce our conclusions and their implications for researchers and practitioners.

## 2. Literature Review

### 2.1. Industrial Clusters

The first approach to clusters can be traced back to the classical theories of location from the 19th century [39]. Nevertheless, the interest in this concept reached its most significant expression in more recent decades [40]. Since the 1990s, the name cluster became the most widespread to describe the phenomenon originating from a firm's agglomeration, either in the case of sectoral specialization or in a regional concentration [41]. With the visibility of clusters, several variations of the concept have emerged: Italian Industrial Clusters [42], Innovative Milieu [43], New Industrial Spaces [44], Industrial Cluster [45], Industrial Location Process [46], Regional Innovative System [47], and Learning Regions [48]. The absence of a single definition has made the concept susceptible to criticism [49], and the lack of clarity has become even worse as the notion is frequently confused with neighbouring concepts used as equivalents or synonyms [50].

One of the most recent approaches to territorial agglomeration is linked to industrial clusters described as "geographic concentration of interconnected companies, suppliers, service providers, firms in related industries, and associated institutions (e.g., universities, standard agencies and trade associations) in particular fields that compete but also cooperate" [45] (p. 197). This approach is compatible with the theoretical perspective of industrial clusters as a construct that aggregates both geographical and network dimensions.

In this way, this article adopts the constructivist perspective of industrial clusters to integrate an actor-centred and structural perspective, focusing on the network dimension [51] to explain the international expansion of clustered firms. Prior research has demonstrated that networks, which act as transmitters of knowledge, are more efficient within industrial clusters [52]. Although the use of a network perspective is not new [53] some research gaps still exist [54]. There have been few studies that explicitly identify and measure these networks [54] and little evidence has been provided about their dynamics, that is, in how they form and change over time [55].

The recognition that local and trans-local linkages are important for clustered firms to acquire knowledge and resources, pushed scholars to go beyond the traditional local–global dichotomy to adopt the network view of industrial clusters [11]. These structures are rarely self-sufficient, and are limited to being considered as isolated systems [56]; in turn, they

correspond to networks of local relationships embedded in a larger "global cluster network" exchange that provides valuable assets [57].

Early studies have shown that, in industrial clusters, geographical proximity is crucial for establishing informal collaboration and knowledge exchange [58]. For firms, co-locating with related companies has the advantage of boosting a collective learning process, enabling the acquisition of resources that otherwise could not be obtained [59]. Regarding the role played by networks, several recent studies suggest that the establishment of such interactions is the key to the success of industrial clusters [11,12,60–62].

Thus, networks are defined as inter-relationships that connect actors with common interests [63], facilitating the development of different ties to obtain mutual benefits [64]. The term is used to denote a set of connected agents [65], which may be organizations, individuals, customers, suppliers, service providers, or government agencies [20]. Recently, Foghani, Mahadi, and Omar [66] conceptualised a network as:

> "*Alliances belonging to group of companies that function together to achieve an economic objective and cooperate based on joint development projects, while complementing on another and specializing to solve common challenges and reaching a collective efficient goal, while conquering markets that would have been too difficult to reach on their own.*" [66] (p. 2)

In light of the above, network relationships have been conceptualised through distinctive perspectives (On the one hand, tie strength is a function of "the amount of time, the emotional intensity, the intimacy (mutual confiding), and the reciprocal services which characterize the tie" [67] (p. 1361). On the other hand, social capital corresponds to "the sum of the actual and potential resources embedded within, available through, and derived from network relationships possessed by an individual or social unit" [68] (p. 243). Finally, tie configuration relates to the differences in the network's location. According to Prashantham and Young [69] "bridging (external) social capital is based primarily in the international market(s)", while "bonding (internal) social capital is likely to spread both domestic base and international market(s)" (p. 281)): (1) Tie Strength [67]; (2) Social Capital [68]; and (3) Tie Configuration [69]. Furthermore, such interactions are distinguished into social and business networks [14]. The term social network is interlinked with informal or interpersonal ties, while business networks are attached to formal or inter-organisational relationships [70]. Regarding the location of their partners, these networks can also be classified as national and international; the former relates to the contacts established with other entities inside the home country, while the latter refers to the international relationships developed by firms [20].

### 2.2. Speed of Internationalisation

The topic of the speed of internationalisation has emerged as an important issue in the international entrepreneurship (IE) literature due to the recent focus on early internationalisation driven by globalisation [71]. Internationalisation is defined as "*the process through which firms increase their exposure and response to international opportunities and threats*" [72] (p. 71). Scanning the literature, it has become quite common to differentiate between the initial speed of entry (earliness) and the speed that one firm reaches after entering foreign markets (post-internationalisation speed) [69,72].

The earliness of internationalisation is usually conceptualised through different expressions used as synonyms: accelerated internationalisation [73], rapid foreign entry [74], internationalisation speed [13,14], speed of entry [75], precocity [18,20], and early internationalisation [4,19,76]. The most relevant studies in this area measure earliness as the amount of elapsed time between the first internationalisation and the firm's founding [4,14,76], or the difference between the year of the firm's inception and the year that the first export was undertaken [13,75].

On the other hand, the dynamics of internationalisation after a firm's first international market has been achieved (post-internationalisation speed), have received little attention in the IE literature [72]. This concept is frequently defined through the change

in international scale, scope, and pace. A usual metric for international scale (also called degree of internationalisation or extent) includes the ratio of foreign sales to total sales (FSTS), indicating the percentage of a firm's sales generated from foreign markets [32,76]. The international scope captures the geographic diversity, representing, for example, the number of countries where firms export [35,73]. Finally, international pace relates to the level of foreign direct investment (FDI) frequently operationalised through the average number of new subsidiaries per year [24,27].

The use of different expressions to describe the speed of internationalisation introduces too much complexity at the conceptual level, since "*research has not sufficiently distinguished between two closely but distinct issues*" [76] (p. 909). Considering the dynamism associated with the speed of the internationalisation process, two theoretical approaches have been used to explain a firm's internationalisation—the Uppsala Model and the International New Venture (INV) Theory. The Uppsala Model considers internationalisation as a gradual commitment to foreign markets [77,78], overlooking competition and strategy dynamics [73]. Conversely, the INV theory focuses on internationalisation as an accelerated process [71], ignoring the time-dependent process of knowledge and competencies [32].

Considering the above arguments, the Network Theory emerges, focusing on experiential knowledge as the key to boosting the internationalisation process [51]. Through cooperation networks, a company gains access to other firms' knowledge without necessarily going through the same experiences; a typical internationalisation process has changed from incremental development to expansion in leaps [51]. Several scholars [71,73] have highlighted that foreign expansion is better understood by integrating both frameworks because each of them focuses on certain dimensions and ignores others.

Both the Uppsala Model and the INV theory display the relevance of a firm's interactions for internationalisation. While the INV theory highlights that established networks are vital for early internationalisation [79], the Uppsala Model considers that many firms enter international markets almost blindly, as it is important to develop several networks to increase their chances of survival [80].

### 2.3. Performance

The literature has stressed that a rapid decision-making process enables firms to exploit international opportunities [81], enhance performance [82], and allows them to achieve a better competitive advantage [83]. Consequently, performance is conceptualised as the achievement of growth while ensuring a firm's survival [26].

Considering the dynamics of international markets, firms are moved by a mix of financial and non-financial motivations and, occasionally, a trade-off between these two dimensions may emerge [84]. Previous research has emphasised that performance can be measured by both financial (e.g., sales growth, export profitability, R&D intensity) and non-financial indicators (e.g., firm's innovation, growth in the number of employees, strengthening strategic position) [1,14,19,25,70]. While financial indicators are objective and clear, they also have some limitations when they are used to compare the performance of several firms with different goals, sizes, industry backgrounds, and strategic visions; in these cases, non-financial performance measures might be more suitable [85]. Regarding this issue, Cavusgil and Zou [86] underlined that considering performance only through sales or profits ignores a firm's strategy and competitive ambitions.

Thus, operationalising this concept is a challenging endeavour [87] since there is a large heterogeneity associated with its indicators. Furthermore, most empirical studies studying firms' performance employed unidimensional measures [2,27,29,88], constraining their analysis. However, nowadays, a new strand of the literature has arisen, focusing on exploring firms' performance through a multidimensional perspective [1,14,70] to obtain more generalisable conclusions about their competitive advantages.

## 3. Research Model and Hypotheses

Previous research has noted the role of networks in the internationalisation process [89,90]. According to Johanson and Vahlne [51], belonging to networks enhances successful internationalisation processes because they provide trust, learning, and opportunities in an environment that facilitates a firm's ability to approach foreign markets [91]. Through network embeddedness, firms may overcome their resource constraints and internationalise in a manner that would not otherwise have been possible [92]. This perspective recognises that firms are not isolated entities, considering them as "*systems of social and industrial relationships encompassing, for example, customers, suppliers, competitors, family and friends*" ([89] (p. 365)). Thus, network resources enable firms to cope with the risks and challenges of foreign markets [64].

In the case of industrial clusters, most studies acknowledge that being a part of a network significantly improves the clustered firms' ability to internationalise [69]. These structures can accelerate internationalisation by promoting the system of relationships between their members [89]. By paying attention to networks and knowledge spillovers, scholars have recognised the relevance of extra-local linkages for industrial clusters [93]. This finding is particularly important for SMEs, where being a part of a network acts as a facilitator of their internationalisation [94]. Moreover, with the spillover effect referring to the impact of seemingly unrelated events on the outcome variable [95], a positive (or negative) spillover effect implies synergistic (or trade-off) relationships and beneficial (or detrimental) roles of cooperation networks on the clustered firms' performance.

Although previous research claims a positive relationship between the network's development and internationalisation, in terms of the speed of internationalisation, some conflicting findings emerge. The results range from a positive influence [5,13,16,33,35,73] to a non-significant effect [14,18–20]. Decisions about the speed of internationalisation become increasingly important for gaining and sustaining competitive advantages [1]. The IB literature emphasises the positive outcomes provided by rapid internationalisation underlining that firms "should internationalise aggressively" to enhance performance [23] (p. 483). However, the speed of internationalisation can be a double-edged sword [1]. Through first-mover advantages [96] and learning advantages of newness [76], accelerated internationalisation can lead firms to success, while internationalising slowly can mean the loss of valuable business opportunities [97]. Nonetheless, due to external liabilities, aggressive internationalisation can endanger a firm's survival. Thus, organisations are challenged to face several risks and foster their growth and performance while, at the same time, being confronted with strong organisational constraints [98].

Overall, the IB literature claims a positive relationship between international expansion and performance [34]. Over the last few years, many empirical studies have explored the relationship between the speed of internationalisation and performance, providing mixed results [1,24,27,29,98]. The current lack of consensus is aggravated by the difficulty of conceptualizing both speeds of internationalisation and performance. Some studies have understood speed as the time until internationalisation starts [19,25,26]; others have focused on the speed of international operations once the firms had expanded abroad [1,24,30,32,34]. Consequently, there is a need to make a further explicit distinction between these two close, but different concepts, to develop more rigorous studies [99].

Considering this issue, Zahra and George [100] observed that the degree of internationalisation, international scope, and speed are the three dimensions of IE that have received the most attention. This study adopts their framework but, due to the scarcity of available data, we only consider two dimensions of the speed of internationalisation: (1) international scale which captures the level of internationalisation that the firm has achieved considering the FSTS growth; and (2) international scope that comprises the number of countries or, in our case, geographic markets in which firms obtain their international sales.

Therefore, the mediating role of the speed of internationalisation in contributing to clustered firms' performance is central in our paper (Figure 1). Depending on the acceptance of hypotheses, the model can be purely or partially mediated. In the first case,

the influence of network clustering on performance will only be mediated by the speed of internationalisation. The second case would entail a direct effect of network clustering on performance, plus an indirect effect through the speed of internationalisation.

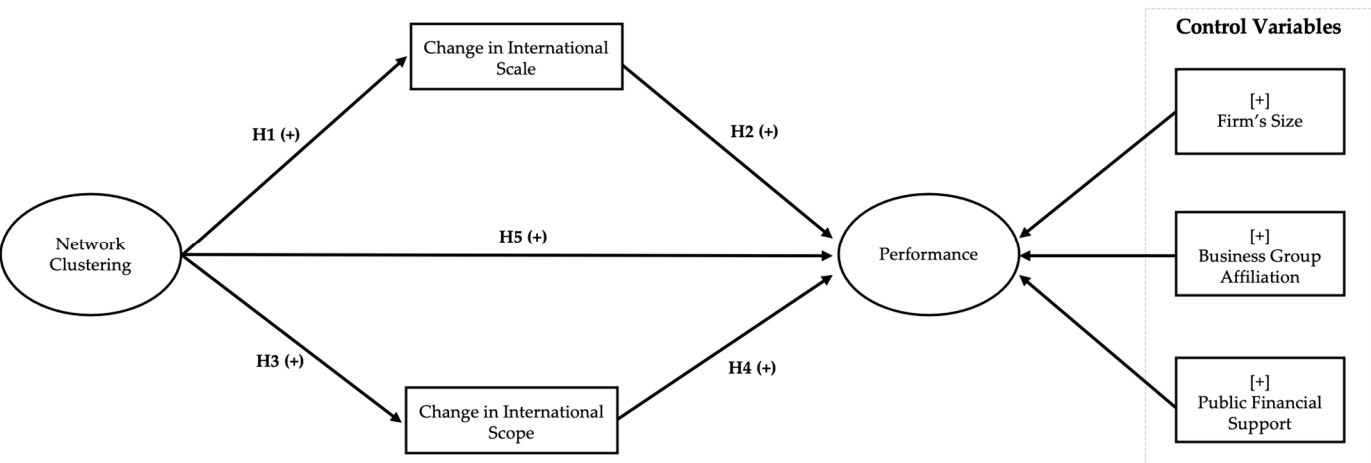

**Figure 1.** Research model. Note: On the one hand, network clustering and performance are the constructs or latent variables (i.e., variables that are not directly measured) represented by circles. These constructs have a measurement model that specifies the relationship between each construct and its indicator variables (i.e., network clustering is measured by national networks and international networks, while performance is measured by sales growth, R&D intensity, and innovation). On the other hand, the speed of internationalisation dimensions—change in international scale and scope—are observed variables represented by rectangles.

International sales (i.e., international scale) are the first dimension of IE and the one that has received the most attention in the literature [100]. Firms internationalise not only to exploit the capabilities developed in their home countries but also to access resources that are not available in those markets [71]. In this way, the interactions established within the firms' network allow them to obtain new experiences, resources, and knowledge, which can have a leverage effect on their ability to enter international markets [101]. According to Crespi, Crisquolo, and Haskel [102], firms that exported in the past are more likely to learn from customers, and those who learned with these sources are in a better position to increase productivity growth, which is consistent with the learning by exporting hypothesis [102–104].

According to Prashantham [15], the firms that use their local networks have a higher probability of increasing their level of export intensity and international competitiveness. In a similar vein, Boehe [33] pointed out that local ties in an industry association strongly influence the level of international sales. Thus, networks play a significant role in promoting and facilitating clustered firms' internationalisation [17].

However, the influence of networks on the several dimensions of the speed of internationalisation is not equal. Pla-Barber and Escribá-Esteve [73] found that the intensity of network relationships with customers and competitors increases the likelihood of achieving a higher percentage of exports and increases the number of countries where the firm sells, accelerating the internationalisation process. Nevertheless, the supplier network increases the likelihood of adopting a slower internationalisation and the linkages established with other institutions are not significant, neither for export levels nor geographic diversity.

Additionally, it has been argued that a firm's international experience contributes to its ability to recognise international opportunities [105]. According to Himersson [34] and Hitt, Hoskisson, and Kim [106] a rapid internationalisation intensity can offer cost-based advantages, a more efficient use of the firm's resources, and the achievement of scale economies leading, eventually, to a higher market share and financial returns [90].

In the IB literature, the role of the international scale on performance has not been consistent. While some studies have found a positive relationship [22,23,32] others have revealed a negative effect [88,107,108], nonlinear [31,106], or even non-significant effects [2,34]. Although conflicting findings persist, some scholars have suggested that a positive relationship between these two dimensions may exist [92].

In light of the above arguments, searching for business opportunities in international markets is a part of network relationships. Clustered firms' that are orientated to develop such relationships will exhibit higher growth in the degree of internationalisation (i.e., change in international scale). Likewise, as the firms' international scale increases, learning opportunities are created exerting a positive impact on performance. These arguments allowed us to formulate the following hypotheses:

**Hypothesis 1.** *For clustered firms, the establishment of network relationships leads to a higher change on an international scale.*

**Hypothesis 2.** *A higher change in international scale has a positive impact on the performance of clustered firms.*

Although the international scale provides information about the firm's foreign expansion, some studies have suggested the use of other measures incorporating greater multidimensionality [73]. Therefore, international scope reflects a second dimension of IE [100], which offers a more fine-grained view of the firm's international strategy [109].

The relationship between networks and the speed of internationalisation has been examined by a large number of papers [4,5,14,20,33,73], but the relationship with international scope has received less attention [35]. Previous research has conceptualided that social networks have a positive effect on two dimensions of speed, namely, international commitment and country scope [69]. This strand of the literature emphasises the importance of social and business networks in accumulating knowledge [110], their capability to influence established partners, and the speed of internationalisation [111].

More recently, Felzensztein, Ciravegna, Robson, and Amorós [35] showed that networks are an important means for firms to support their internationalisation strategies, especially when they are targeting markets outside the domestic country. Similarly, the entrepreneur's experiences suggest that having a higher number of networks leads to a more diverse internationalisation. The benefits of acquiring strategic resources, beyond national borders, are more pronounced for firms expanding into multiple countries [112]. In this way, a higher geographical diversity increases the likelihood of internationalised firms obtaining critical resources, enabling them to catch up with the competition and improve their performance [113].

The relationship between geographical diversification and a firm's performance also has a long history. As Contractor, Kundu, and Hsu [28] contended "the foundation of international business studies rests on the assumption that increased multinational is good for a firm performance" (p. 5). Several scholars have supported that international scope positively influences a firm's performance [2,22,32,34]. However, other studies such as Chang [29] and Collins [108] have found a negative relationship, while Sadeghi, Rose, and Chetty [1] demonstrated a nonlinear effect.

Despite the conflicting findings, it is believed that operating in multiple regions, even when using low-commitment entry modes, exposes firms to new realities, providing a platform that enables access to different sources of knowledge [92,106]. Accordingly, to explore new international markets, clustered firms should intensively use their networks to overcome resource constraints. In turn, a broader international scope enhances knowledge acquisition and mitigates the liabilities of newness and foreignness, allowing one to attain better performance. Consistent with empirical explanations, we theorise that:

**Hypothesis 3.** *For clustered firms, the establishment of network relationships leads to a higher change in international scope.*

**Hypothesis 4.** *A higher change in international scope has a positive impact on the performance of clustered firms.*

Based on the social capital theory, IB researchers have emphasised the relevance of networks to enhance firms' performance [114,115]. By working together and exchanging information, firms can share the risks of failure and trepidation intertwined with the internationalisation process [116]. In this regard, Faria, Lima, and Santos [117] also highlighted that cooperative firms have, on average, higher performance levels than non-cooperative firms since they can share investment costs and may take advantage of partners' resources and capabilities.

In light of the above, previous research has reinforced the role of managerial and social networks on strategic choices and performance [115,118]. The managers' ties with other firms and government agencies help to improve business performance in terms of market share and return on assets [119]. Moreover, the information and resources exchanged within the personal network are believed to improve financial indicators, such as revenue and profitability [114]. Accordingly, Yeoh [120] showed that personal sources of information and social connections with other network individuals positively influence the export performance of international SMEs.

More recently, Musteen, Francis, and Datta [14] explored the influence of international networks on the speed of internationalisation and performance. Their findings suggested that firms that share a common language with their partners internationalise faster, and those having diversified international relationships display superior performance. Hence, the IB literature recognises that firms can use cooperation networks to capture business opportunities in foreign markets, overcome internationalisation barriers, and improve competitive advantage [121].

Consistent with previous research, it is expected that firms embedded in industrial clusters will be able to improve their learning process and resource acquisition, reflecting that ability in higher levels of performance. Thus:

**Hypothesis 5.** *For clustered firms, the establishment of network relationships leads to higher performance.*

## 4. Methodology

### 4.1. Data Sources

The criteria and rationale for selecting the population and variables of our empirical tests are explained in the following paragraphs. The first step involved identifying the Portuguese industrial clusters. For this purpose, we consulted the Agency for Competitiveness and Innovation (IAPMEI) website [37], which allowed us to obtain a total of 19 clusters. Then, the cluster managing organizations were contacted to provide the following information: (1) classification of the clustered firms' economic activities (NACE codes) (NACE is the abbreviation from Nomenclature statistique des activités économiques dans la Communauté européenne and represents the European standard classification of productive economic activities. Particularly, the CIS database provides the NACE Rev. 3 classification implemented in 2007. For more information on the NACE classification, please refer to: https://www.ine.pt/ine_novidades/semin/cae/CAE_REV_3.pdf (accessed 27 July 2022)), (2) geographical location of the cluster, (3) identification of the firms and other organizations (e.g., universities, research centres, public authorities, among others) formally (According to the cluster managing associations, to be considered a cluster member, the firms must fulfil the following criteria: (a) identify themselves with the purposes of the cluster, (b) exhibit the NACE codes required by the managing organizations, and (c) pay the membership fee) associated to the cluster, and (4) membership conditions. The initial contact was made via email and, later, by telephone to reinforce the request for participation in the study

conducted between October 2019 and February 2020. At the end of data collection, 17 valid answers were obtained (response rate = 89.5%).

In the second step, the statistical data were gathered using the responses to three (The survey questions used to collect quantitative data in the CIS database were the following: (1) "Please indicate the economic activities developed by clustered firms (NACE codes)", (2) "Are the clustered firms concentrated in any regions of the national territory? Or are they more dispersed?", (3) "What is the geographical location of affiliated firms?") questions from the survey sent to the cluster managing associations, and the CIS database [36] was selected to collect quantitative information. Analysing the 17 responses, we concluded that only 10 provided all information requested. However, 6 did not match the firm's NACE codes available on the CIS database [36] and, for this reason, they were excluded. Thus, our analysis focused on four (Only 4 of the 19 Portuguese industrial clusters were considered to fulfill the specific requirements for the analysis, introducing some bias in the sample. If it was possible to include all industrial clusters, we would have had access to a greater number of firms formally belonging to these structures, enabling us to overcome this issue. However, despite the efforts to obtain all information requested, some managing cluster associations did not participate in the survey due to several reasons. Some were unreachable after multiple attempts, and others declined to provide specific data or indicated a lack of time to participate.) clusters: (1) Footwear and Fashion; (2) Textile—Technology and Fashion; (3) Petrochemical, Industrial Chemistry and Refining; and (4) Automotive. Our sample, therefore, included Portuguese firms derived from manufacturing sectors—footwear, textile, chemical and automotive—considering the NACE codes provided by the cluster's managing entities.

Since our dependent variable was the firms' performance measured through indicators, such as R&D intensity and innovation, the CIS instrument appeared as a reliable data source, allowing the use of R&D intensity as the input of innovation and product/process/organisational/marketing innovations as outputs of innovation [122]. This survey aimed to collect data on innovation understood from a broader perspective rather than exclusively examining the invention process. Thus, the CIS questionnaire comprised a wide range of innovation activities going beyond R&D expenditures, personnel training, market analysis, and trial production to include the introduction of innovative production processes and organisational changes [117]. Following the Eurostat recommendations, the Portuguese version directly collects information on cooperation partners, degree of internationalisation, geographical markets in which the firm sells, R&D intensity, as well as product, process, organisational, and marketing innovations.

The dataset included the timeframe between 2012 and 2014, considering firms with ten or more employees operating in different sectors. The CIS questionnaire was available between 9 October 2014 and 8 June 2016 [123]. Based on a census combination for large firms and random sampling for other groups, the survey consisted of 9455 enterprises. In the corrected sample of 8736 companies, 7083 valid answers were considered (i.e., 81% response rate) [123]. Since our purpose was to evaluate how the changes in international scale and scope help clustered firms to increase their performance, based on the inter-organisational relationships established within industrial clusters, our sample included the enterprises that could belong to the four aforementioned clusters. Moreover, as the research focus was also on the speed of the internationalisation process, we selected the firms that had at least one year of international sales. At the date of data extraction (June 2020), 1491 firms met all the above criteria (Table 1).

Table 2 provides the means, standard deviations, and Pearson correlation coefficients. As shown in the table, the correlations between the variables of interest were relatively low, suggesting that multicollinearity did not affect our results. Regarding common method bias (CMB), which is a potential problem when the variables are obtained from the same data source [124], we used two procedures to control and detect CMB. First, we used Harman's single-factor test [125] as an exploratory approach. To do so, we forced all items used in this study to load on one single factor. This test resulted in a 26.17% variance explained. As this

factor did not account for the majority of covariance between the measures, we assumed that CMB was not a pervasive issue in this study [126]. An extraction with eigenvalues above 1 with Varimax rotation confirmed this interpretation as all items loaded highly on their respective scales. Second, following Kock [127], we conducted a test based on collinearity assessment. This procedure aims to assess if the variance inflation factors (VIFs) are above 3.3, indicating pathological collinearity in the data. We analysed all VIF values in the partial regressions and found that they were clearly below the cut-off value of 3.3. Hence, this result was consistent with the one produced by Harman's single-factor test, that is, we concluded that there was no evidence of CMB in this study.

**Table 1.** Selection criteria based on CIS [33].

| CIS [33] | Number of Firms (n) |
|---|---|
| Census combination and random sampling | 9455 |
| Corrected sample | 8736 |
| Valid answers | 7083 |
| Firms displaying the NACE codes required by the four cluster-managing associations | 2884 |
| Firms that had, at least, one year of international sales | 1491 |
| Sample representativeness based on the population of 7083 enterprises | 21.05% |

**Table 2.** Descriptive statistics.

| | Mean | S.D. | Min | Max | 1 | 2 | 3 | 4 | 5 | 6 | 7 |
|---|---|---|---|---|---|---|---|---|---|---|---|
| 1. Business Group Affiliation | 0.305 | 0.461 | 0 | 1 | 1 | | | | | | |
| 2. Firm's Size | 2.000 | 0.500 | 1 | 3 | 0.308 *** | 1 | | | | | |
| 3. Change in International Scale | 0.434 | 2.700 | −1.000 | 42.000 | 0.045 (+) | 0.052 ** | 1 | | | | |
| 4. Change in International Scope | 1.816 | 0.389 | 1 | 2 | 0.059 (+) | 0.165 *** | 0.044 (+) | 1 | | | |
| 5. Network Clustering | 1.801 | 1.786 | 0 | 19 | 0.116 *** | 0.103 ** | −0.017 | 0.067 *** | 1 | | |
| 6. Performance | 0.933 | 1.262 | −0.151 | 1.691 | 0.117 *** | 0.086 (+) | 0.020 | 0.133 *** | 0.248 ** | 1 | |
| 7. Public Financial Support | 0.857 | 0.765 | 0 | 3 | 0.089 ** | 0.188 *** | 0.033 | 0.126 *** | 0.162 ** | 0.209 *** | 1 |

Mean, standard deviation (S.D.), minimum (min), and maximum (max) values. $p$-values significant at (+) $p < 0.05$, ** $p < 0.005$, *** $p < 0.001$.

### 4.2. Variables and Statistical Procedure

A structural equation model was used to test the hypotheses in SmartPLS software 3.2.9 [128]. The PLS-SEM was primarily selected because:

1.  This method works efficiently when used to estimate path models with many indicators, constructs, and relationships [129,130];
2.  The PLS-SEM supports both explanatory and predictive goals when testing the model's causal-predictive relationships [131];
3.  The technique performs well when using secondary data and larger sample sizes [129,130];
4.  This method allows us to account for and estimate the effect of mediating variables [132];
5.  Our sample displays some distribution issues, such as a lack of normality (The lack of distributional assumption was one of the main reasons for choosing PLS-SEM. However, it is worth noting that in a limited number of situations non-normal data

may also influence PLS-SEM results [133]. The use of bias-corrected and accelerated (BCa) bootstrapping handles these issues, as it adjusts the confidence intervals for skewness [134]. Following this guideline, we employed BCa bootstrapping to correct the data for both bias and skewness.) [129];

The CIS questionnaire was divided into thirteen sections. There were two sections assessing internationalisation activities, one section for national and international networks, and four sections accounting for product, process, organisational, and marketing innovations. To evaluate firms' performance (dependent variable), we adopted financial and non-financial indicators, relying on a multidimensional approach [1]. According to several contributions from the literature, financial performance was assessed through sales growth [25,32,107]. In addition, we also included R&D intensity [106] operationalised through R&D spending as a percentage of total sales, because it is frequently used as a measure of innovative activities [135] and demonstrates the importance of the interrelation between the knowledge creation processes and the exporting activities [103]. With regards to non-financial performance, it was measured considering firms' innovation [25], including a typology of product, process, organisational, and marketing innovations [136]. Following the extant literature [137], innovation was understood as the development of something new (radical innovation) and/or the gradual improvement of something that already exists (incremental innovation). Accordingly, it was argued that both innovations were not mutually exclusive and may be used as complementary actions to deal with external demand. For this research, we used any type of innovation (radical or incremental);

With regards to network clustering (independent variable), to identify the entities that may belong to industrial clusters, we adopted the NACE codes provided by the cluster management organizations [94,138]. Focusing on the network dimension of industrial clusters [139], the inter-organisational relationships were operationalised considering two dimensions: (1) national networks, embracing the relationships developed on the domestic market, and (2) international markets, representing the interactions outside the home-country [14,20]. In our study, "network relationships" were defined as the firms' interactions with other companies of the same group, customers, suppliers, competitors, consultants/commercial labs, universities/other higher education institutes, government, and public/private research institutes;

Based on Zahra and George [100], the speed of internationalisation (mediating variable) was measured by the changes registered in international scale and scope. Following previous studies [140], we measured the change in international scale with the following formula: $\frac{(Foreign\ Sales_{2014}/Total\ Sales_{2014}) - (Foreign\ Sales_{2012}/Total\ Sales_{2012})}{(Foreign\ Sales_{2012}/Total\ Sales_{2012})}$, reflecting the FSTS growth between 2012 and 2014. On the other hand, the change in international scope reflects the geographic markets where the firms' sales are generated [35,141], proxied by whether the firm sells for European Union (EU) and/or non-EU markets (The EU markets include EU members and associated countries: Albania, Germany, Belgium, Bosnia-Herzegovina, Bulgaria, Cyprus, Croatia, Denmark, Slovakia, Slovenia, Spain, Estonia, Finland, France, Greece, Hungary, Ireland, Iceland, Italy, Kosovo, Latvia, Liechtenstein, Lithuania, Luxembourg, Macedonia, Malta, Montenegro, Norway, Netherlands, Poland, United Kingdom, Czech Republic, Romania, Serbia, Sweden, Switzerland, and Turkey. The non-EU markets embrace the United States, China/India, and other countries around the world.) during the timeframe under analysis;

Finally, to deal with unobserved heterogeneity, we controlled the effects of some variables potentially affecting firms' performance. This study, therefore, controlled for the firm's size (number of employees), business group affiliation (dummy variable), and public financial support (incentives/tax benefits, subsidies, loans, or bank guarantees that one firm has received). To summarise, Appendix A provides complete information about the measurement of the variables, and how they relate to the CIS questionnaire (see Table A1).

## 5. Results

### 5.1. Data Adequacy

We tested our hypotheses using the PLS-SEM since the goal was to maximise the explanation of variance ($R^2$) for clustered firms' performance in a latent model. According to Chin [142], this procedure is more robust than a variance-covariance-based model for small and medium-sized samples. The first concern relates to the minimum sample size required to evaluate relationships. The widely used rule of thumb [142] suggests that the overall sample size should be 10 times the largest of: (1) the block with the larger number of indicators or (2) the dependent variable with the largest independent variables impacting it. In our model, (1) is equal to 3 (performance) and (2) is equal to 6 (the number of arrows arriving at performance). Therefore, the minimum sample size should be 60 and our sample contained 1491 cases, so data adequacy was met.

In our framework, network clustering and performance were the constructs or latent variables (i.e., variables that are not directly measured). These constructs have a measurement model that specifies the relationship between each construct and its indicator variables (i.e., network clustering is measured by national and international networks, while performance is measured by sales growth, R&D intensity, and innovation). The dataset had only 100 missing values, which were coded with the value $-99$. The maximum number of missing data points per item was 18 of 1491 (1.21%) in the firm's size. Since the relative number of missing values was very small, we continued the analysis by using the mean value replacement of the missing data option. Box plot diagnostic using IBM SPSS statistics software 28 [143] revealed influential observations, but no outliers. This evidence, therefore, allowed us to proceed with model estimation.

### 5.2. Measurement Checks

Exploratory factor analysis was conducted to assess the reliability and validity of the latent variables using IBM SPSS statistics software 28. The results of the exploratory factor analysis are presented in Table 3. The measure of the adequacy of the Kaiser–Meyer–Oklin (KMO) compares simple correlations with partial correlations. Our output resulted in a KMO of 0.606 meetings the KMO criteria between 0.50 and 1 [144]. Furthermore, Bartlett's sphericity test verified that the correlation matrix was an identity matrix which would imply that its intercorrelations were zero. This test took a value of 114,361 (10 d.f.) with a *p*-value below the significance level of 0.001. This means that the observed variables were correlated, justifying the use of factor analysis.

**Table 3.** Exploratory factor analysis.

| Latent Variables | Observed Variables | MSA (Anti-Image Matrix) | Communalities Extracted | Total Variance Explained | Component Matrix | KMO and Bartlett's Test |
|---|---|---|---|---|---|---|
| Network Clustering | National Networks | 0.634 | 0.585 | 37.295% | 0.748 | KMO = 0.606 Approx. chi-square = 114,361 df = 10 Sig < 0.001 |
| | International Networks | 0.601 | 0.668 | | 0.783 | |
| Performance | Firm's Innovation | 0.653 | 0.492 | | 0.665 | |
| | R&D Intensity | 0.568 | 0.918 | 58.543% | 0.955 | |
| | Sales Growth | 0.501 | 0.264 | | 0.492 | |

The variable changes in international scale, changes in international scope, firm's size, business group affiliation, and public financial support are not included in the analysis because they are single items.

On the other hand, the diagonal of the anti-image matrix contained the measures of sample adequacy (MSA), comparing the magnitude of the coefficients of the observed variables with the magnitude of the coefficients of the partial correlations, in which all variables must reveal MSA values above 0.50 [145]. Since none of the observed variables had MSA values below 0.50, it was not necessary to remove any of them. The communalities extracted, representing the amount of total variance of the original variables explained by

the common factors (i.e., high communalities indicate the amount of variance that was extracted by the factors), returned values above 0.50 for most variables [145]. Only the observed variable—sales growth—showed less common variability than the others (less than 0.50); however, it was maintained in the analysis because its MSA value was slightly above 0.50 (MSA = 0.501: Table 3). The total variance explained also met the criteria of being higher than 0.50 [145].

Finally, the extraction was based on the principal component method with an eigen-value greater than 1 and maximum iterations for convergence equal to 25 (unrotated factor solution). This method of extraction is adequate when the objective is to summarise most of the original information (variance) in a minimum number of factors, with prediction purposes [145]. Moreover, the Varimax method was applied with maximum iterations for convergence equal to 25. After the extraction, two factors emerged corresponding to the reflective latent variables:

- Factor 1—Network Clustering: constituted by the observed variables of national networks and international networks;
- Factor 2—Performance: composed of the observed variables of sales growth and R&D intensity (financial performance) and the firm's innovation (non-financial performance).

Upon the identification of which observed variables constituted the latent variables through exploratory factor analysis, the following step was carried out in SmartPLS software 3.2.9 adopting a rule that retained observed variables must meet the minimum threshold of 0.60 [146]. Since this confirmatory factor analysis was related to the evaluation of the reflective measurement models, a detailed explanation of this step can be found in the following subsection.

### 5.3. Reflective Outer Model Evaluation

After running the algorithm, the evaluation of the PLS-SEM results began with the assessment of the reflective measurement models (i.e., network clustering and performance). Table 4 shows the results and evaluation criteria outcomes. In the case of reflectively measured constructs, we should start by examining the indicator loadings (i.e., outer loadings). According to Hair, Hult, Ringle, and Sarstedt [146], loadings above 0.60 indicate a sufficient level of reliability. Since all outer loadings ranged between 0.681 and 0.898, they exceeded the recommended threshold.

**Table 4.** Assessment of convergent validity and internal consistency reliability.

| Constructs | Indicators | Convergent Validity | | | Internal Reliability | | |
|---|---|---|---|---|---|---|---|
| | | Loadings | Reliability | AVE | CR $\rho_c$ | $P_A$ | CA |
| Network Clustering | National Networks | 0.785 | 0.616 | 0.711 | 0.830 | 0.748 | 0.730 |
| | International Networks | 0.898 | 0.806 | | | | |
| Performance | Firm's Innovation | 0.681 | 0.464 | 0.529 | 0.618 | 0.557 | 0.543 |
| | R&D Intensity | 0.772 | 0.596 | | | | |

AVE, average extracted variance; CR, composite reliability; CA, Cronbach's alpha. The variables change in international scale, change in international scope, firm's size, business group affiliation, and public financial support are not included in the analysis because they are single items. The indicator sales growth was removed from the measurement model relative to performance because returned an outer loading equal to 0.174, clearly, below the recommended threshold of 0.60.

Next, we analysed the convergent validity of the latent variables. The convergent validity measures the extent to which a construct converges in the indicators by explaining the items' variance [130]. Thus, the convergent validity was assessed by the average variance extracted (AVE) for all indicators associated with a construct. An acceptable AVE should be 0.50 or higher, since this indicates that, on average, the construct explains over 50% of the variance of its items [130]. The AVE for network clustering was 0.711 and for performance corresponded to 0.529, revealing convergent validity [147].

The next step involved the assessment of the constructs' internal consistency reliability. When using PLS-SEM, internal consistency reliability is typically evaluated using the composite reliability $\rho_c$ (CR), where higher levels of values indicate greater levels of reliability. According to Hair, Risher, Sarstedt, and Ringle [129], CR values between 0.60 and 0.70 are considered acceptable in exploratory research, and values between 0.70 and 0.90 range from satisfactory to good. All CR values (ranging from 0.618 to 0.830) were higher than the suggested cut-off value of 0.60. On the other hand, Cronbach's alpha (CA) is another measure of internal consistency reliability that assumes similar thresholds but produces lower levels than CR [129]. Specifically, CA is a less precise measure of reliability as the items are unweighted. Conversely, in CR the items are weighted based on their loadings and, thus, the indicators' reliability is higher than in CA [129]. The CA value for network clustering fulfilled the recommended threshold of 0.60 in explanatory research, whereas the CA values of performance were slightly lower (CA. = 0.543) [129].

While CA may be too conservative, the CR can be too liberal, and the construct's true reliability is typically viewed as within these two extreme values. As an alternative, Dijkstra and Henseler [148] proposed $\rho_A$ as an approximately exact measure of construct reliability, which usually lies between CA and CR. In our case, the $\rho_A$ of network clustering met the recommended threshold of 0.707 [148], while performance did not ($\rho_A = 0.557$). However, considering the explanatory nature of this research, the lower values of CA and $\rho_A$, and the acceptable levels of AVE and CR $\rho_c$ for performance, allowed proceeding with the analysis [149].

Once the reliability and convergent validity of the reflective constructs were established, the next step involved assessing the discriminant validity (Table 5). According to Sarstedt, Ringle, Smith, Reams, and Hair [130], discriminant validity shows the extent to which a construct is empirically distinct from other constructs in the path model. The most conservative technique to assess discriminant validity is the Fornell and Larcker [147] criterion. This method compares each squared root of AVE values with the correlations between the latent variables. In our sample, the correlations between each pair of constructs did not exceed the square root of AVE [147].

**Table 5.** Assessment of discriminant validity.

| | Fornell and Larcker Criterion | | HTMT Correlation | |
|---|---|---|---|---|
| | **1** | **2** | | **Network Clustering** |
| 1. Network Clustering | *0.843* | | | |
| 2. Performance | 0.248 | *0.727* | Performance | 0.418 [0.307; 0.574] |

The italic numbers on the diagonal are the square root of AVE. Off-diagonal value is the correlation between both latent variables. The values in the brackets represent the 95% confidence intervals. The variable changes in international scale, change in international scope, firm's size, business group affiliation, and public financial support are not included in the analysis because they are single items.

However, recent research has highlighted that this criterion is not the most reliable for the assessment of discriminant validity. Henseler, Ringle, and Sarstedt [150] showed that the Fornell and Larcker technique does not perform well when the indicator loadings vary slightly (i.e., when they range between 0.65 and 0.85). Based on this limitation, these scholars proposed the hetero-trait mono-trait (HTMT) of the correlations, which compares the mean value of items correlations with the (geometric) mean of the average correlations for the item measuring the same latent variable [129]. For conceptually distinct variables, Henseler et al. [150] recommended a conservative threshold of 0.85 for the HTMT correlations. Additionally, the bootstrapping procedure can also be applied to test whether the confidence intervals of the HTMT correlations do not include the value of 1 [150].

In our sample, we concluded that the HTMT correlation for the relationship between network clustering and performance was below the cut-off value of 0.85. We also ran the bootstrapping with 5000 subsamples choosing the bias-corrected and accelerated (BCa)

bootstrapping and the one-tailed testing at a 5% significance level. The results revealed that the HTMT correlation fell within the 95% confidence intervals and, those intervals did not include the value of 1, suggesting that discriminant validity was established between the pair of constructs. The reflective measurement models, therefore, indicated that the measures displayed satisfactory levels of reliability and validity, allowing us to proceed to the structural model evaluation.

### 5.4. Inner Model Evaluation

The second step of the PLS-SEM analysis involved the assessment of the structural model based on three parameters: (1) the relevance and significance of path coefficients, (2) the in-sample explanatory power ($R^2$ and $f^2$), and (3) the out-of-sample predictive power ($Q^2$). Furthermore, before this evaluation, the structural model must be assessed for potential collinearity in the partial regressions [130].

The estimation of the path coefficients was obtained through multiple regression analyses. Therefore, it was extremely important to ascertain whether the regression results were not biased by collinearity issues. Since all VIF values were below the recommended threshold of 5 [151] (Table 6), we concluded that multicollinearity was not a problem. Then, the strength and significance of the path coefficients were examined through bootstrapping as the basis for estimating t-values [130]. We report the results of the path coefficients in Tables 6 and 7. In our model, the path coefficients ranged from −0.017 to 0.214 with different significance levels. The meaning of these coefficients is further discussed in the following section.

**Table 6.** Bootstrap analysis and statistical significance of direct effects.

| | Performance | | | Change in International Scale | | | Change in International Scope | | |
|---|---|---|---|---|---|---|---|---|---|
| | $\beta$ | VIF | $f^2$ | $\beta$ | VIF | $f^2$ | $\beta$ | VIF | $f^2$ |
| Business Group Affiliation | 0.075 (2.580 *) | 1.116 | 0.006 | | | | | | |
| Firm's Size | −0.004 (0.105 ns) | 1.163 | <0.001 | | | | | | |
| Change in International Scale | 0.011 (0.366 ns) | 1.006 | <0.001 | | | | | | |
| Change in International Scope | 0.095 (2.612 *) | 1.040 | 0.010 | | | | | | |
| Public Financial Support | 0.157 (2.940 **) | 1.069 | 0.026 | | | | | | |
| Network Clustering | 0.208 (2.330 +) | 1.043 | 0.046 | −0.017 (0.960 ns) | 1.000 | <0.001 | 0.067 (4.099 ***) | 1.000 | 0.005 |
| $R^2$ | | 0.106 | | | | | | | |
| SRMR | | | 0.045 [a] | | | | | 0.046 [b] | |
| Critical Thresholds: at 95% | | | 0.052 | | | | | 0.052 | |
| at 99% | | | 0.056 | | | | | 0.055 | |

VIF, inner VIF values for the partial least regressions; R2, explained variance; f2, effect size. [a] The saturated model represents the correlations between all the constructs. [b] The estimated model is based on a total effect scheme (i.e., it considers the model structure depicted). Path coefficients significant at *p*-values: [+] $p < 0.050$; [*] $p < 0.010$; [**] $p < 0.005$; [***] $p < 0.001$. The values in the brackets represent t-values. t-values thresholds at one-tailed test of alpha = 0.05 and 5000 resamples: t (0.05; 4999) = 1.645; t (0.01; 4999) = 2.327; t (0.005; 4999) = 2.576; t (0.001; 4999) = 3.091.

**Table 7.** Bootstrap analysis and statistical significance of indirect effects.

| | Estimate ($\beta$) | Lower Bounds (BC) | Upper Bounds (BC) | *p*-Value |
|---|---|---|---|---|
| Specific indirect effects Network Clustering $\rightarrow$ Change in International Scale $\rightarrow$ Performance | 0.000 (0.267 [ns]) | −0.003 | 0.000 | 0.395 |
| Network Clustering $\rightarrow$ Change in International Scope $\rightarrow$ Performance | 0.006 (2.286 [+]) | 0.003 | 0.012 | 0.011 |
| Total indirect effects Network Clustering $\rightarrow$ Performance | 0.006 (2.239 [+]) | 0.002 | 0.011 | 0.013 |
| Total effects (indirect plus path) Network Clustering $\rightarrow$ Performance | 0.214 (2.416 [*]) | 0.072 | 0.374 | 0.008 |

BC, bias-corrected. Path coefficients significant at *p*-values: [+] $p < 0.050$; [*] $p < 0.010$. The values in the brackets represent t-values. t-values thresholds at one-tailed test of alpha = 0.05 and 5000 resamples: t (0.05; 4999) = 1.645; t (0.01; 4999) = 2.327; t (0.005; 4999) = 2.576; t (0.001; 4999) = 3.091.

The next step involved reviewing the in-sample explanatory power ($R^2$ and $f^2$). The $R^2$ is a measure of the variance explained in the dependent variable accounting for the model's predictive accuracy. The $R^2$ value for performance (i.e., target construct) was 10.6% exceeding the acceptable cut-off point of 10% [152]. This meant that 10.6% of the variability observed in the clustered firms' performance was explained by the variables included in the structural model. Moreover, the effect size ($f^2$) complemented the $R^2$ assessment, considering the relative impact of an independent variable on the dependent one through the changes in $R^2$ values [153]. According to Cohen [153], the $f^2$ effect size can be classified as follows: $f^2 \geq 0.35$ (high), $0.15 \leq f^2 < 0.35$ (medium), $0.02 \leq f^2 < 0.15$ (small), and $f^2 \leq 0.02$ (negligible). Overall, our $f^2$ effect sizes were mostly classified as small or negligible (Table 6).

The final step required the assessment of the out-of-sample predictive power ($Q^2$). The $Q^2$ is based on the blindfolding procedure, which omits a part of the data matrix, therefore estimating the model parameters and predicting the omitted part by using the previously computed estimates [130]. This analysis focused on the dependent variable and its indicators. We determined the predicted relevance by performing the blindfolding procedure using an omission distance of seven (D = 7) [130]. Table 8 shows that the indicators of performance achieved $Q^2$ values above zero, indicating that the model outperformed the naïve benchmark [133]. To classify the model's predictive power, we ran the PLS$_{predict}$ with ten folds and ten repetitions [154]. The prediction errors produced by the PLS path models showed that the distribution was not highly unsymmetric. Hence, the analysis focused on the root-mean-square error (RMSE) statistics (Table 8). Since the RMSE values produced by the PLS-SEM were consistently lower than the one of the linear models (LM) benchmark, we concluded that the model revealed a high out-of-sample predictive power [154].

**Table 8.** PLS$_{predict}$ results.

| | | RMSE | |
|---|---|---|---|
| Indicators | $Q^2$ Predict | PLS-SEM | LM |
| Firm's Innovation | 0.094 | 0.252 | 0.278 |
| R&D Intensity | 0.122 | 0.530 | 0.567 |

$Q^2$ predict, cross-validated redundancy; RMSE, root-mean-square error; PLS-SEM, PLS path model; LM, linear model benchmark.

To complete the validation of the model, the overall fit (Model fit indices enable a judgement of how well a hypothesised model structure fits the empirical data. Nevertheless, the notion of model fit known from covariance-based structural equation modelling (CB-

SEM) is not transferable to PLS-SEM as the method follows a different aim when estimating model parameters (the aim is to maximise the explained variance rather than minimise the divergence between covariance matrices) [129]. Yet, research has brought forward several PLS-SEM-based model fit measures, such as the standardised root mean square (SRMR), RMStheta, and the exact fit test [155] which, however, have proven ineffective in detecting model misspecifications in settings usually encountered in applied research. Instead of assessing model fit, the structural model assessment in PLS-SEM focuses on evaluating the model's explanatory and predictive power [151].) of PLS-SEM was evaluated using the standardised root mean square residual (SRMR) [156]. Both saturated and estimated models displayed an SRMR value below the recommended threshold of 0.08 [155], being smaller than their corresponding 95% and 99% quantiles [157].

*5.5. Robustness Tests*

To check the validity of the findings, further analysis was conducted. First, we checked the existence of potential nonlinearities because PLS-SEM assumes linear relationships by default [158]. We therefore applied Ramsey's [159] test in the latent variables scores extracted after the convergence of the PLS-SEM algorithm. The results revealed that the partial regressions of the independent variables on performance (F $(4; 1465 = 0.429$, $p = 0.921$), as well as on the change in international scale (F $(2; 1464 = 1.521$, $p = 0.304$) and on the change in international scope (F $(2; 1464 = 0.581$, $p = 0.615$), were not subject to nonlinearities. Hence, we concluded that the linear effects model was robust.

Second, since the speed of internationalisation (i.e., change in international scope) has a positive effect on performance, but the reverse can also happen (Gkypali, Tsekouras, and von Tunzelmann [160] highlighted the existence of potential endogeneity between technological and foreign market forces, represented by R&D (firm's innovation) and export intensity (internationalization.)), that is, the firm performance may also have an impact on the speed of internationalisation, we checked our model for potential endogeneity. Our assessment of potential endogeneity followed the Hult, Hair, Proksch, Sarstedt, Pinkwart, and Ringle [161] approach, starting with the application of Park and Gupta's [162] Gaussian copulas, using the latent variable scores of the original model. The first step consisted of verifying whether the variables were non-normally distributed resorting to the Kolmogorov–Smirnov test [143]. The results showed that none of the variables had normally distributed scores, allowing us to proceed with Park and Gupta's [162] procedure. Considering the independent variables as potentially endogenous, they revealed non-significant coefficients for all potential combinations of Gaussian copulas (i.e., the *p*-values were higher than the significance level of 5%). We thus concluded that endogeneity was not a problem in our data [161]. For the sake of clarity, the results of the robustness tests are available in Appendix B (see Tables A2 and A3).

## 6. Discussion

Our results supported some of the hypotheses of this study (Table 6, Figure 2). The relationship between network clustering and the speed of internationalisation was partially confirmed. First, we found no support for hypothesis 1, which proposed that, for clustered firms, the establishment of network relationships led to a higher change in international scale (H1: $\beta = -0.017$, $p = 0.169$). One possible explanation for this result is that, in our study, 86.3% of the clustered firms are SMEs. (According to the EU recommendation 2003/36, SMEs are defined as firms with less than 250 employees and annual sales below EUR 50 million or annual balances below EUR 43 million). SMEs face significant challenges in obtaining foreign knowledge and overseas contacts, encountering several obstacles when going abroad [97]. Smaller firms are confronted with more barriers than their larger counterparts in building business relationships since, from the firm's perspective, the establishment of networks corresponds to an intensive investment [163]. Due to resource constraints, SMEs are self-reliant and operate in isolation [164], being mostly reactive to "serendipities" [165]. Thus, these firms are given to inertia in network interactions, being

more reluctant to trust their potential partners which, in turn, could help them to spread their sales in international markets.

**Figure 2.** Results of direct effects.

In contrast, hypothesis 3—for clustered firms, the establishment of network relationships leads to a higher change in international scope—was supported (H3: $\beta$ = 0.067, $p < 0.001$). This finding implies that firms developing national and international interorganisational relationships, through several economic activities rooted in industrial clusters, display a higher geographical diversity, which is consistent with existing studies claiming a positive relationship between the establishment of networks and internationalisation scope [35,69,112]. Nowadays, firms find themselves competing internationally, regardless of their size [166]. In an attempt to grow in global markets, the ability to leverage social and business networks has become crucial [167]. Another strand of the literature on SMEs networking emphasised their proactive behaviour in pursuing foreign business development [168]. The extant literature has discussed the importance of networks in supporting and enhancing SMEs' internationalisation [89], showing that these relationships are important vehicles to explore international opportunities. In this way, networks are particularly important for SMEs' international expansion [169], as they provide alternative paths to entry into new international markets, help them to evaluate potential partners, and reduce exchange risks [170]. Therefore, the benefits of network interactions are more pronounced for firms expanding into multiple countries [112], so the development of network interactions in industrial clusters allows them to achieve a higher international scope.

On the other hand, the effect of the speed of internationalisation on firms' performance was also partially observed. Hypothesis 2—a higher change in international scale has a positive impact on the performance of clustered firms—did not receive support (H2: $\beta$ = 0.011, $p$ = 0.357). Hence, the change in international scale has no effect on clustered firms' performance, corroborating some of the findings reported in the IB literature about the relationship between the degree of internationalisation (i.e., scale) and performance [2,34]. Previous studies [51] have argued that an increased international scale would lead to a greater foreign outlook, that enables warning signals to be identified and recognised, resulting in a better performance. However, our results seemed to contradict this tenet, at least for firms in industrial clusters. One reason for this might be that the international scale—measured as the ratio between foreign sales and total sales—provides a very narrow view of the clustered firms' internationalisation process [87]. Despite the FSTS ratio being the most widely used measure for international scale in IB studies [1,2,76], this

operationalisation reveals some limitations [87], which have been methodologically raised by Certo, Busenbark, Kalm, and LePine [171]. First, it is a ratio, increasing the probability to be affected by modifications in both numerator (foreign sales) and denominator (national plus international sales), that is, there is a possibility that the changes in FSTS are only triggered by domestic sales. Second, it may reflect the internationalisation of distinctive value chain stages. Third, ratios can inflate standard errors reducing the likelihood of finding statistical coefficients when, in fact, there are effects to observe. They may also reduce statistical power and effect sizes. Our results, therefore, could have been influenced by using the FSTS ratio to measure the change on an international scale.

However, it is important to note that rapid growth in the foreign market is, in many cases, the only way for firms (especially SMEs) to survive in highly competitive international markets. Considering their scarce resources and the existence of a trade-off between the resources to be allocated to explore and exploit activities, that is, the firm cannot allocate more resources to both activities, but can only increase that commitment by detracting resources from the other activity, can justify a lower R&D intensity (non-financial performance). Hence, according to the exploitation perspective, the learning based on the existing experience and resources, besides guaranteeing a low risk, allows an immediate growth of company sales when expanding its operations abroad associated with deepening value delivery within an existing customer [172].

Conversely, hypothesis 4—a higher change in international scope has a positive impact on the performance of clustered firms—was supported (H4: $\beta$ = 0.095, $p < 0.01$). This means that clustered firms selling on different geographical markets display better performance. The present study, therefore, confirms early insights in the IB literature [2,32,34] claiming that a broader scope exposes firms to a multitude of institutional environments, allowing them to transform those experiences into knowledge and experiential learning, to spread risks, and to balance sales fluctuations between different markets, improving their performance [22]. Furthermore, penetrating more than one geographically distinct market allows firms to charge premium prices for their products, thus, spreading their costs and expanding the appropriate returns over innovation investments [173]. Hence, we found that firms in industrial clusters will benefit from the synergies developed within the cluster ecosystem—e.g., access to past experiences, learning, and knowledge of other cluster members—allowing them to enjoy the several advantages brought by developing their activities in different worldwide regions, particularly, on achieving a better performance. In this case, access to external know-how by different geographical markets may leverage the efficiency of internal R&D activities, at least if a firm is willing to accept external ideas and knowledge, overcoming the "not invented here" syndrome [174].

Likewise, hypothesis 5—for clustered firms, the establishment of network relationships leads to higher performance—was also validated (H5: $\beta$ = 0.208, $p < 0.05$). These outcomes suggest that the development of national and international networks through industrial clusters enhances clustered firms' performance [14,114,120]. Previous research has largely agreed that networks influenced performance outcomes, such as market entry, selection and growth, by exposing the firms to new knowledge, business opportunities, and additional networks [16]. Some studies even proved that vertical relationships are important drivers of a firm's decision to engage in R&D both independently and with external partners [175]. Thus, firms embedded in industrial clusters should be aware of the potential benefits resulting from their networks that can support the achievement of better performance, i.e., innovation is no longer the province of individual firms but depends increasingly on collective action [176].

Furthermore, our study provides additional insights. The results indicated that network clustering has a direct and significant impact on clustered firms' performance ($\beta$ = 0.208, $p < 0.05$: Table 6), but also an indirect, mediated effect, through the change in international scope ($\beta$ = 0.006, $p < 0.05$: Table 7). However, it is worth noting that the indirect relationship between network clustering and clustered firms' performance, mediated by the change in international scale, was not statistically significant ($\beta$ = 0.000, $p = 0.395$:

Table 7). Given this scenario, we conclude that our model is partially mediated only by one dimension of the speed of internationalisation (direct effect of network clustering on performance plus an indirect effect through the change in international scope), suggesting that a combination of network clustering and geographical diversity (scope), helps to improve firms' performance in industrial clusters.

Regarding control variables, business group affiliation has a positive impact on clustered firms' performance ($\beta = 0.075$; $p < 0.01$). Thus, agglomerated firms belonging to a business group are in a better position to improve their performance. This can be explained by the fact that, in these groups, individual firms share multiple links (e.g., cross-ownership and close market ties), allowing them to achieve mutually recognised goals [177]. In addition, public financial support is positively related to performance ($\beta = 0.157$; $p < 0.005$), that is, clustered firms that have received incentives/tax benefits, subsidies, loans or bank guarantees from public institutions, can perform better. Indeed, subsidizing private R&D and innovative capabilities, help to overcome financial constraints and foster economic growth [178], resulting in greater performance. Finally, in our sample, the effect of the firm's size on performance was not statistically significant ($\beta = -0.004$; $p = 0.458$).

## 7. Conclusions

This paper contributes to a better understanding of the clustered firms' performance, considering the role of network relationships and the speed of internationalisation in industrial clusters. The inclusion of network clustering in the IB literature is still scarce [11,12], and the combination of both topics—network clustering and speed of internationalisation—can yield additional insights to answer the question of how clustered firms can speed up their international expansion and achieve superior performance. Early studies returned doubts about the nature of the relationship between both network clustering → speed of internationalisation [4,18,19], and speed of internationalisation → performance [1,27,112]. Thus, we tried to answer this question by introducing the mediating role of the speed of internationalisation on the relationship between network clustering (cooperation networks) and clustered firms' performance. In doing so, we found that the change in international scope is one of the channels through which the networks established in industrial clusters lead to improved performance, while the change in international scale did not produce the same result.

Our study has several theoretical and practical implications. From a theoretical point of view, we contribute to the convergent efforts to link regional and international business fields. While the IB literature has traditionally overlooked the regional context in which the economic activity of the firm takes place, the interaction between firms and territory is steadily emerging as the missing piece for understanding the speed of internationalisation and performance. Thus, we attempted to address this research gap by introducing the role of industrial clusters to account for the existence of physical and social–institutional closeness with other cluster members that can, eventually, influence the clustered firms' speed of internationalisation and performance outcomes. We found that a portion of the performance improvements is rooted in the networks established through the cluster, and this progress is partially explained by the change in the diversity of geographical markets (scope). A rapid international expansion can be a source of competitive advantage for firms in industrial clusters. To remain competitive and ensure strategic positioning, particularly when competing in a dynamic environment (such as the international markets), firms should focus on developing and maintaining close relationships with several partners that provide privileged information about new foreign countries. Thus, the resources that are needed to boost a faster and more successful internationalisation process, which contributes to improved short-term performance, are available in industrial clusters due to the cooperation networks established between different actors.

These findings also have important implications for practitioners by highlighting the need to carefully consider the speed at which firms spread their sales. Through network clustering, managers of internationally clustered firms can diversify the risks between

different countries, to reduce sales fluctuations and gain flexibility. This happens because the other counterparts already established in the cluster might support and complement the profound knowledge and experience of the newly clustered firms' owners and managers, thereby improving their strategic roles. This research has shown that the change in international scope catalyses both financial and non-financial aspects of performance. In this way, rapid internationalisation is a relevant weapon for clustered firms that should be properly managed, because faster may not always be better. Managers should be aware of the complexities and potential effects of rapid international growth. Specifically, in clustered SMEs that face financial constraints and limited international experience, managers need to be cautious when deciding to speed up the level of geographical diversification to avoid harming performance.

This study also has practical implications for policymakers. From a policy perspective, our results revealed that the exchange of export-related information must be encouraged. In fact, since firms' international behaviour can be influenced by the spillover effect triggered by the cluster atmosphere, export-promotion initiatives (e.g., trade fairs), and regional public agencies aimed at coordinating local firms' internationalisation, could prove useful in fostering clustered firms' participation in international activities. In light of this study's findings, there might be room for policymakers to focus on removing, or at least mitigating, some of the impediments to internationalisation activity, such as the high level of indebtedness and diseconomies of scale due to firms' limited size. Moreover, any public intervention requires specific policies and actions that need to take into consideration the type of actors that constitute the regional structure, and their interactions with the geographical space. Policies known as "one-size-fits-all", trying to boost regional internationalisation, have several limitations because not all clustered firms act in the same way due to different objectives influencing their behaviour. Hence, when tailoring policy interventions, policymakers should account for the heterogeneity of the economic actors belonging to industrial clusters.

While this article offers interesting findings, we are aware of its limitations. First, this study is limited in scope since we only tested a sample of Portuguese firms. Although the findings can be generalised to a limited extent to other small, open, and relatively well-developed countries, future research should expand the analysis to other contexts to account for distinctive institutional and cultural settings. Since emerging countries and their firms are constantly aiming to catch up, or reduce the gap, with the more advanced economies [179], future studies could focus on explaining the catch-up processes in the context of clustered firms by comparing those of emerging and advanced economies. Moreover, according to Porto, Lee, and Mani [179], macroeconomic variables—such as wage rate, exchange rate, and FDI—are important explanatory variables to understand the rise, fall, and re-rise of firms; thus, those macroeconomic variables might be added to our research model to capture the "whole picture" at the cluster level. Second, the database used has two main limitations: (1) the CIS data are usually available for the community for a lot of time after being collected, and (2) the dataset is limited to three years (2012–2014). As pointed by Faria, Lima, and Santos [117], the cross-sectional nature of the CIS data can be mitigated by developing a panel data study to complement the findings and conclusions presented in this paper. Third, our measure of the change in international scale is captured by the FSTS ratio, which might have influenced the study's results. Based on the limitations of the FSTS ratio as a measure of international scale [87,171], a more fine-grained operationalization of this dimension is warranted for future studies to analyse its potential mediation in the relationship between network clustering and performance. Fourth, alternative measures for cluster affiliation should be used to investigate whether clustered firms' performance is sensitive to other operationalizations of industrial clusters. Specifically, complementary measures such as the location quotient [180]—widely used in regional studies to characterise industrial specialization [181]—can be applied in future research efforts. In this regard, emerging approaches also analyse industrial clusters as a system of interconnected enterprises and an element of multi-level policymaking. These approaches emphasise

public participation and a continuous redefinition of sustainability challenges in response to changes in socio-ecological systems [182]. In this context, the concept of integrated sustainability emerges to reflect the urgency of managing the complex and unpredictable nature of socio-ecological systems and their multiple stakeholders, acknowledging that various issues are simultaneously related to local contexts and larger external systems. Integrated sustainability, as a contemporary and worldwide spread concept, is considered the most dominant pillar of sustainable development when compared to the traditional ones (social, environmental, and economic), implicitly, claiming that policymakers should follow flow-based governance (rather than place-based governance), and establish international agreements to foster the countries openness [95]. Although integrated sustainability indicators for industrial clusters address clustered firms as an element of both economic and socio-ecological systems and as actively integrated in multi-level policymaking and planning [182], due to constraints on data collected, we were unable to include and measure this concept in our study. Hence, future research could focus on the interconnectedness between integrated sustainability indicators and clustered firms' performance as a dynamic process rather than an end goal.

On the other hand, in our study, clustered firms' performance was measured by a combination of financial performance (sales growth and R&D intensity) and non-financial performance (firm's innovation); however, the reflective model evaluation has reduced the measurement of performance to innovative variables (We would like to thank the anonymous reviewer for this important observation which allowed us to develop contextualised future research avenues) (R&D intensity and firm's innovation: Table 4). Thus, our dependent variable was represented by the clustered firms' innovativeness, unveiling three potential avenues for further research. First, the relationship between cooperation networks and innovation is not straightforward. From a subjective perspective (i.e., firm-based), most enterprises still develop their new products/services without forming (formal) cooperative arrangements; yet, the empirical evidence has shown that firms engaging in R&D and attempting to introduce a high level of innovation—i.e., "new to the market" rather than "new to the firm" innovations—are much more likely to engage in cooperation networks [117,175,176] and, consequently, produce a higher innovative output [183,184]. Accordingly, future studies should focus on exploring how the clustered firms' performance is sensitive to the degree of complementarity between cooperation networks and innovation. Second, although our findings provided evidence of non-reverse causality between the speed of internationalisation and innovation-based performance, several studies highlight the relevance of a firm's innovation to increasing the degree of internationalisation [185,186]. This research stream is particularly focused on testing the learning by exporting hypothesis [102–104]; thus, another avenue for further research could be the development of frameworks applying this baseline assumption in the cluster context. Third, alternative measures for clustered firms' performance—such as growth [122], survival [135], and economic performance [187]—might be used to assess the robustness of this study's results.

Finally, the clustered networks-performance "nexus" stands out as a promising opportunity for investigation with qualitative research methods (e.g., in-depth interviews, and case studies). Through an in-depth investigation of the economic and socio-spatial dynamics in context, it would be possible to understand the genius loci of clustered firms, assessing how the sense of place is produced and evolves, uniquely influencing the relationships between clustered firms, speed of internationalisation, and performance.

**Author Contributions:** Conceptualisation, C.S. and T.M.; methodology, T.M.; software, T.M.; validation, A.B. and C.S.; funding acquisition, A.B., C.S. and T.M.; writing—original draft preparation, T.M.; writing—review and editing—A.B. and C.S. All authors have read and agreed to the published version of the manuscript.

**Funding:** This work has been supported by national funds through FCT—Fundação para a Ciência e Tecnologia under the projects UIDB/04728/2020 and UIDP/04728/2020.

**Institutional Review Board Statement:** Not applicable.

**Informed Consent Statement:** Informed consent was obtained from all subjects involved in the study.

**Data Availability Statement:** The data presented in this study are available on request from the corresponding author. The data are not publicly available due to privacy restrictions.

**Conflicts of Interest:** The authors declare no conflict of interest.

## Appendix A

**Table A1.** Description of variables.

| Variables | | Measurement | Theoretical Foundation | Proxy | Source |
|---|---|---|---|---|---|
| Dependent Variable | Performance | Financial Performance | e.g., [25,32,107] | Sales growth = (Total Sales$_{2014}$ − Total Sales$_{2012}$)/Total Sales$_{2012}$ | (Question 14.1; CIS [33]) What was your enterprise's total turnover for 2012 and 2014? (Expressed in thousands of euros) |
| | | | e.g., [106] | R&D Intensity = (External R&D + Internal R&D)/Total Sales$_{2014}$ | (Question 5.2; CIS [33]) How much did your enterprise spend on in-house R&D and external R&D in 2014? (Expressed in thousands of euros) |
| | | Non-Financial Performance [1] | e.g., [25,136,137] | Product Innovation: sum of all 2 items (ordinal variable ranging from 0 = the firm does not innovate in good/service to 2 = the firm innovates in both) | (Question 2.1; CIS [33]) During the three years 2012–2014, did your enterprise introduce (1) goods innovation, and (2) service innovations? (Dummy variables: 0 = no, 1 = yes) |
| | | | | Process Innovation: sum of all 3 items (ordinal variable ranging from 0 = the firm dos does not innovate in the process to 3 = the firm innovates in all items of process innovation) | (Question 3.1; CIS [33]) During the three years 2012–2014, did your enterprise introduce new or significantly improved (1) manufacturing methods, (2) logistics, delivery, or distribution methods, and (3) supporting activities? (Dummy variables: 0 = no, 1 = yes) |
| | | | | Organisational Innovation: sum of all 3 items (ordinal variable ranging from 0 = the firm dos does not innovate in organisational methods to 3 = the firm innovates in all items of organisational methods) | (Question 8.1; CIS [33]) During the three years 2012–2014, did your enterprise introduce (1) business practices, (2) organizing work responsibilities and decision-making, (3) organizing external relations? (Dummy variables: 0 = no, 1 = yes) |
| | | | | Marketing Innovation: sum of all 4 items (ordinal variable ranging from 0 = the firm dos does not innovate in marketing to 4 = the firm innovates in all items of marketing) | (Question 9.1; CIS [33]) During the three years 2012–2014, did your enterprise introduce new methods of (1) designs or packaging (2) product production, (3) product placement, and (4) pricing? (Dummy variables: 0 = no, 1 = yes) |
| Independent Variable | Network Clustering | Cluster Affiliation | e.g., [94,138] | NACE codes to identify the firms that may belong to industrial clusters | IAPMEI [34] Cluster Managing Associations (Question 1; CIS [33]) General information about the firm: main activity (two-digit NACE codes) |
| | | National Networks | e.g., [14,20] | National Networks: sum of all 8 items for each partner in Portugal (ordinal variable ranging from 0 = the firm does not collaborate with any partner in Portugal to 8 = the firm collaborates with all types of partners in Portugal) | (Question 7.2; CIS [33]) Please indicate the type of cooperation partner by location: (1) other enterprises in the same group, (2) suppliers, (3) customers from the private sector, (4) customers from the public sector, (5) competitors, (6) consultants, (7) universities, (8) government/ research institutes (Dummy variables: 0 = no, 1 = yes) |
| | | International Networks | | International Networks: sum of all 32 items for each partner in European countries, the US, China/India, and all other countries (ordinal variables ranging from 0 = the firm does not collaborate with any partner in foreign countries to 32 = the firm collaborates with all types of partners in foreign countries) | |

**Table A1.** *Cont.*

| Variables | | Measurement | Theoretical Foundation | Proxy | Source |
|---|---|---|---|---|---|
| Mediating Variable | Speed of Internationalisation | Change in International Scale | e.g., [100,140] | Change in International Scale = $(FSTS_{2014} - FSTS_{2012})/FSTS_{2012}$ | (Question 14.2; CIS [33]) What was the percentage of year total turnover from sales to clients outside your country in 2012 and 2014? (Expressed as a percentage) |
| | | Change in International Scope | e.g., [35,100,141] | Change in International Scope: sum of all 2 items (ordinal variable ranging from 1 = the firm sells to one geographical market to 2 = the firm sells for both markets) | (Question 1.3; CIS [33]) In which geographic markets did your enterprise sell goods and/or services during the three years 2012 to 2014? (Dummy variables: 0 = no, 1 = yes) |
| Control Variables | Firm's Size | Number of employees | e.g., [26,34] | Firm's Size: ordinal variable (coded as 1 = under 50 employees, 2 = 50–249 employees, 3 = over 250 employees) | (Question 14.3; CIS [33]) What was your enterprise's average number of employees in 2012 and 2014? |
| | Business Group Affiliation | The firm is a part of a business group | e.g., [24,177] | Business Group Affiliation: dummy variable (1 = if the firm belongs to a business group, 0 = otherwise) | (Question 1.1; CIS [33]) In 2014, was your enterprise part of an enterprise group? (Dummy variables: 0 = no, 1 = yes) |
| | Public Financial Support | Financial support for innovation | e.g., [178] | Public Financial Support: sum of 3 items (ordinal variable ranging from 0 = the firm does not receive financial support from any public entity to 3 = the firm received financial support from all public entities) | (Question 6.1; CIS [33]) During the three years 2012–2014, did your enterprise receive any public financial support for innovation activities from the following levels of government: (1) local or regional authorities, (2) central government, (3) European Union (EU)? (Dummy variables: 0 = no, 1 = yes) |

[1] Non-financial performance measured by the firm's innovation is a composite variable computed as the sum of product, process, organisational, and marketing innovation (ordinal variable ranging from 0 = the firm does not innovate in any item of product, process, organisational, and marketing innovation to 12 = the firm innovates in all items of product, process, organisational, and marketing innovation).

## Appendix B

**Table A2.** Assessment of nonlinear effects.

| Nonlinear Relationships | Coefficient | *p*-Value | $f^2$ | Ramsey's Test |
|---|---|---|---|---|
| Network Clustering × Network Clustering → Performance | 0.025 | 0.379 | 0.001 | |
| Change in International Scale × Change in International Scale → Performance | 0.040 | 0.218 | 0.004 | $F_{(4; 1465)} = 0.429$, $p = 0.921$ |
| Change in International Scope × Change in International Scope → Performance | 0.037 | 0.491 | 0.003 | |
| Firm's Size × Firm's Size → Performance | 0.014 | 0.769 | 0.000 | |
| Business Group Affiliation × Business Group Affiliation → Performance | 0.045 | 0.290 | 0.005 | |
| Public Financial Support × Public Financial Support → Performance | 0.071 | 0.135 | 0.008 | |
| Network Clustering × Network Clustering → Change in International Scale | 0.031 | 0.259 | 0.006 | $F_{(2; 1464)} = 1.521$, $p = 0.304$ |
| Network Clustering × Network Clustering → Change in International Scope | 0.052 | 0.423 | 0.009 | $F_{(2; 1464)} = 0.581$, $p = 0.615$ |

**Table A3.** Assessment of endogeneity.

| Models | Independent Variables | Coefficient | *p*-Value |
| --- | --- | --- | --- |
| Gaussian copulas of the performance model (endogenous variables: network clustering, change in international scale, change in international scope, firm's size, business group affiliation, public financial support) | Network Clustering | 0.006 | 0.774 |
| | Change in International Scale [c] | 0.028 | 0.876 |
| | Change in International Scope [c] | 0.021 | 0.293 |
| | Firm's Size [c] | 0.039 | 0.832 |
| | Business Group Affiliation [c] | 0.022 | 0.273 |
| | Public Financial Support [c] | 0.012 | 0.752 |
| Gaussian copulas of the change in international scale model (endogenous variables: network clustering) | Network Clustering [c] | 0.026 | 0.243 |
| Gaussian copulas of the change in international scope model (endogenous variables: network clustering) | Network Clustering [c] | 0.101 | 0.351 |

[c] indicates the Gaussian copulas in the models. Each of the models includes all the predictor variables.

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
