# Peer review of "Dancing with Giants: A Unified Framework for Cooperation Networks, Speed of Internationalisation, and Performance"

_sustainability, doi:10.3390/su15032003_

Round 1
Reviewer 1 Report
This manuscript is a result of hardworking but has some limitations.
This manuscript lack a clear objective and novelty. I recommend the author to add these elements into the introduction.
Table 4 should name the variable at the left hand-side.
I suggest the author to add the concept of "integrated sustainability" into the manuscript in the introduction. This perspective lays a foundation for readers understand a general outlook of this research framework.
Table 7 shows the value of R2 which is unclear and irrational.
The conclusion section should be transformed into a general analysis to propose a summary and comprehension of the whole research without expressing details.
Reviewer 2 Report
This is a very well-crafted paper dealing with a relevant issue, that is the possible impact of industrial clustering over internationalization and eventually firm’s performance. Results show that industrial clustering positively affects performance both directly and indirectly, through the increasing geographical scope of exports (while a simple increasing in export intensity does not turn out to be significant). The adopted methodology is standard (PLS-SEM), but adequate and all statistical checks and robustness tests are well performed.
My main concern is the following: in this study performance is measured by a combination of sales growth and innovation (firm’s innovation and R&D intensity), eventually reduced to the sole innovative variables, since sales growth does not pass the relevant test (see Table 5, on p.13). Therefore, the dependent variable is basically innovativeness; moreover, the adopted database is CIS (Community Innovation Survey), specifically designed to capture firm’s innovative activities. Given this empirical setting, the Authors should radically extend their theoretical setting and literature review (Sections 1 and 2), discussing in deep the relationship between clustering and innovation and between innovation and export (that is also affected by reverse causation through “learning by export). In general terms, the location and motivation of the paper should shift towards the Economics and Management of Innovation both in the Introduction and through an additional sub-section within Section 2. Here below some articles that should be discussed with some detail.
-Barrios, S., Gorg, H. and Stroble E. (2003). “Explaining firms' export behaviour: R&D, spillovers and the destination market”, Oxford B Econ Stat., 65, 475-496.
-Cassiman, B. andVeugelers, R. (2006). “In search of complementarity in innovation strategy: internal R&D and external knowledge acquisition”, Manage Sci, 52, 62-82.
-Crespi, G. Crisquolo, C. and Haskel, J. (2008). “Productivity, exporting, and the learning-by-exporting hypothesis: direct evidence from UK firms”, Can J Economics, 41, 619-638
-Dalgıç, B., FazlıoÄŸlu, B. (2021). Innovation and firm growth: Turkish manufacturing and services SMEs. Eurasian Bus Rev 11, 395–419
-De Faria P., Lima F. and Santos R. (2010). “Cooperation in innovation activities: The importance of partners”, Res Pol, 39, 1082–1092.
-Ganotakis, P. and Love, J. (2011). “R&D, product innovation, and exporting: evidence from UK new technology based firms”, Oxford Econ Pap, 63, 279-306.
-Gkypali, A., Tsekouras K. and von Tunzelmann, N. (2012). “Endogeneity between internationalization knowledge creation of global R&D leader firms: An econometric approach using Scoreboard data”, Ind Corp Change, 21, 731-762
-Gkypali, Areti & Arvanitis, Spyros & Tsekouras, Kostas, 2018. "Absorptive capacity, exporting activities, innovation openness and innovation performance: A SEM approach towards a unifying framework," Technological Forecasting and Social Change, Elsevier, vol. 132(C), pages 143-155.
-Iammarino, S., Piva, M., Vivarelli, M. and Tunzelman, N. (2012). “Technological capabilities and patterns of innovative cooperation of firms in the UK regions”, Reg Stud, 46, 1283-1301.
-Love, J. and Ganotakis, P. (2013). “Learning by exporting: lessons from high technology SMEs”, Int Bus Rev, 22, 1-17
-Piga, C. - Vivarelli, M. (2003), Sample Selection in Estimating the Determinants of Cooperative R&D, Applied Economics Letters, 10, 243-246.
-Porto, T.C., Lee, K. & Mani, S. (2021). The US–Ireland–India in the catch-up cycles in IT services: MNCs, indigenous capabilities and the roles of macroeconomic variables. Eurasian Bus Rev 11, 59–82
-Rebelo, F. and Silva, E. G. (2017). “Export variety, technological content and economic performance: the case of Portugal”, Ind Corp Change, 26, 443–465
-Tether, B. (2002). "Who cooperates for innovation, and why. An empirical analysis”, Res Pol, 31, 947-967.
-Zhang, M., Mohnen, P. (2022). R&D, innovation and firm survival in Chinese manufacturing, 2000–2006. Eurasian Bus Rev 12, 59–95
Reviewer 3 Report
Very happy to read this manuscript, the authors did a very well job. Literature review in this paper is sufficient, and the method was described well. I think this paper could be taken as reference for Portugal enterprise and government. Hence, I suggest to publish this paper.
Round 2
Reviewer 1 Report
Despite long responses of the author, this manuscript maintains its own limitations.
In general, I recommend the author to be more concise regarding manuscript length, response to reviewers, reference list, abstract, and even title.
The English language and punctuation of the manuscript needs revision. One example is that captions do not need point because they are title not sentence.
Table 4 shows no variable!
The concept of "integrated sustainability" and "spillover effects" are beneficial for this manuscript, and the following paper can help the author Sustainability spillover effects and partnership between East Asia & Pacific versus North America: interactions of social, environment and economy.
The value of R2 is irrational and unclear in Table 7.
This manuscript needs a certain, clear, and main objective.
Author Response
Dear reviewer,
Please see the attachment.
Sincerely Yours,
The authors

Reviewer 2 Report
I appreciate the improvements; however, the paper is still not well rooted in the relevant literature and some important references are still missing; in particular, the authors should discuss in detail the following articles:
-Dalgıç, B., FazlıoÄŸlu, B. (2021). Innovation and firm growth: Turkish manufacturing and services SMEs. Eurasian Bus Rev 11, 395–419
-De Faria P., Lima F. and Santos R. (2010). “Cooperation in innovation activities: The importance of partners”, Res Pol, 39, 1082–1092.
-Gkypali, Areti & Arvanitis, Spyros & Tsekouras, Kostas, 2018. "Absorptive capacity, exporting activities, innovation openness and innovation performance: A SEM approach towards a unifying framework," Technological Forecasting and Social Change, Elsevier, vol. 132(C), pages 143-155.
-Piga, C. - Vivarelli, M. (2003), Sample Selection in Estimating the Determinants of Cooperative R&D, Applied Economics Letters, 10, 243-246.
-Porto, T.C., Lee, K. & Mani, S. (2021). The US–Ireland–India in the catch-up cycles in IT services: MNCs, indigenous capabilities and the roles of macroeconomic variables. Eurasian Bus Rev 11, 59–82
Author Response

(The authors gave the same response as above.)

Round 3
Reviewer 1 Report
The revised manuscript has appropriately considered the comments, improved fundamentally. This manuscript is advised to recheck the abstract and the English and academic language of whole the text. The abstract frequently uses "we"! This can be replaced by "this manuscript", "this paper", or "this study". Also, this manuscript needs a deep revision of English language for whole the sections.
Author Response

(The authors gave the same response as above.)

Reviewer 2 Report
I am now happy about the second revision
Author Response

(The authors gave the same response as above.)
